# ZEROTH ORDER OPTIMIZATION BY A MIXTURE OF EVOLUTION STRATEGIES

## ABSTRACT

Evolution strategies or zeroth-order optimization algorithms have become popular in some areas of optimization and machine learning where only the oracle of function value evaluations is available. The central idea in the design of the algorithms is by querying function values of some perturbed points in the neighborhood of the current update and constructing a pseudo-gradient using the function values. In recent years, there is a growing interest in developing new ways of perturbation. Though the new perturbation methods are well motivating, most of them are criticized for lack of convergence guarantees even when the underlying function is convex. Perhaps the only methods that enjoy convergence guarantees are the ones that sample the perturbed points uniformly from a unit sphere or from a multivariate Gaussian distribution with an isotropic covariance. In this work, we tackle the non-convergence issue and propose sampling perturbed points from a mixture of distributions. Experiments show that our proposed method can identify the best perturbation scheme for the convergence and might also help to leverage the complementariness of different perturbation schemes.

## 1 INTRODUCTION

We consider optimizing a function $f(\cdot) : \mathbb{R}^d \to \mathbb{R}$ in the setting that only querying function values is allowed and there is no access to gradients of the function. Alternatively, we aim at minimizing a black-box function in which only the oracle of function value evaluations is available,

$$\min_{\boldsymbol{w}} f(\boldsymbol{w}). \tag{1}$$

There are growing interest in studying "gradient-free" optimization due to its applications in hyper-parameter search (e.g., Bergstra et al. (2011); Koch et al. (2018)), reinforcement learning (e.g., Sehnke et al. (2010); Mania et al. (2018); Salimans et al. (2017); Choromanski et al. (2018); Vemula et al. (2019)), or black-box adversarial attacks on deep neural nets (e.g., Chen et al. (2017); Ilyas et al. (2018); Papernot et al. (2017)). In hyper-parameter search, the goal is to find the best values of a set of hyper-parameters for a machine learning model (e.g., a neural net). One can model the task of hyper-parameter search as optimizing an unknown/black box function as (1). Specifically, one can let $\boldsymbol{w}$ be in the space of hyper-parameters in the way that each element of $\boldsymbol{w}$ represents the value of a specific hyper-parameter and different points $\boldsymbol{w}$ in the hyper-parameter space correspond to different realizations of the hyper-parameters. The mapping (i.e., $f(\cdot)$) from hyper-parameter values to a performance metric (e.g., classification error) of using those hyper-parameter values is unknown. Therefore, the gradient of the mapping/function is not available. One can only query the function value (i.e., obtaining $f(\boldsymbol{w})$) by training a model with the hyper-parameter values indicated by $\boldsymbol{w}$. Obtaining the $\arg\min$ of (1) in this case corresponds to finding the best values of the hyper-parameters (and consequently getting the best trained model). Because of the broad applicability of zeroth order optimization algorithms, recently there has been a spate of research in improving them from different respects.

Standard zeroth order algorithms construct pseudo-gradients by sampling some perturbed points from a Gaussian distribution with an isotropic covariance (e.g., Nesterov & Spokoiny (2017); Duchi et al. (2012) or uniformly from a unit sphere (e.g., Flaxman et al. (2005); Duchi et al. (2012); Shamir (2017)). Algorithm 1 describes a popular technique called "'Gaussian smoothing". It first samples some $K$ perturbed vectors $\boldsymbol{v}_t^k$ from a Gaussian distribution, then a pseudo-gradient $\boldsymbol{g}_t$ is constructed

---

**Algorithm 1** Gaussian smoothing.

---
1: **Require:** hyper-parameters $\sigma$, $K$ and $\eta$.
2: **Init:** $\boldsymbol{w}_0 \in \mathbb{R}^d$.
3: **for** $t = 0$ to $T - 1$ **do**
4:      Sample $K$ perturbed vectors $\boldsymbol{v}_t^k \sim \mathcal{N}(0, \boldsymbol{I}_d)$ for each $k \in [K]$.
5:      Construct a pseudo-gradient $\boldsymbol{g}_t := \sum_{k=1}^{K} \frac{1}{\sigma} \big( f(\boldsymbol{w}_t + \sigma \boldsymbol{v}_t^k) - f(\boldsymbol{w}_t) \big) \boldsymbol{v}_t^k$.
6:      Obtain the next update $\boldsymbol{w}_{t+1} = \boldsymbol{w}_t - \eta \boldsymbol{g}_t$.
7: **end for**

---

by taking a weighted average of these perturbed points. The pseudo gradient is subsequently used to obtain the next update $\boldsymbol{w}_{t+1}$. Very recently, there has been a trend of studies in proposing sampling the perturbed vectors from some non-isotropic Gaussian distributions (e.g. Maheswaranathan et al. (2019); Choromanski et al. (2019a); Ye et al. (2019)). They consider sampling perturbed vectors by $\boldsymbol{v}_t^k \sim \mathcal{N}(0, \boldsymbol{\Sigma})$ such that the covariance $\boldsymbol{\Sigma}$ may not be a scale of the identity matrix. Maheswaranathan et al. (2019) propose letting $\boldsymbol{\Sigma}$ be related with the span of latest $r$ pseudo-gradients to reflect the local geometry. Choromanski et al. (2019a) propose updating $\boldsymbol{\Sigma}$ in the sense to tracking the low-dimensional manifold of the gradient space. Ye et al. (2019) propose letting $\boldsymbol{\Sigma}$ be the inverse of a diagonal matrix in a way that the update rule of each diagonal element is in the fashion of updating the second moment quantity in "Adam" optimization algorithm of Kingma & Ba (2015). Nevertheless, these sampling methods have not really been shown to lead to the convergence to a global minimum even when the underlying function $f(\cdot)$ is convex, albeit some motivating toy examples or some analysis (but not related to the convergence) are provided.

In this paper, we tackle the issue by proposing sampling from a mixture of distributions in which one of them is the standard Gaussian distribution with an isotropic covariance. Our method leverages a bandit algorithm for adaptively selecting which distribution the perturbed vectors are sampled from. By building the theoretical guarantee of the bandit algorithm and the convergence guarantee of the algorithm which always sampling from an isotropic Gaussian distribution (i.e., Algorithm 1), we can show that our method converges with a high probability when the underlying function is convex. We conduct experiments to show that our algorithm works well in practice. Our proposed algorithm is competitive with any algorithms that always sample perturbed vectors from a fixed component (distribution) of the mixture which our algorithm considers. This shows the effectiveness of the proposed method, as one would not know which perturbation scheme is the best beforehand. Furthermore, the proposed algorithm might outperform any of them when the complementariness of different perturbation schemes are exploited.

## 2 RELATED WORKS

Zeroth order optimization is studied in different areas, from optimization, online learning, to bioinformatics. In optimization community, the setting that an algorithm can only query function values might date back to Nemirovski & Yudin (1983) (Chapter 9.3). Nemirovski & Yudin (1983) develop sampling perturbed vectors on a 2-norm sphere for constructing the pseudo gradients. Furthermore, they also develop a different family of zeroth order optimization algorithms that uses binary search (see also Spall (2003); Agarwal et al. (2011); Jamieson et al. (2012)). On the other hand, some lower bound results regarding the iteration complexity are obtained in recent years (e.g., Duchi et al. (2015); Shamir (2013); Jamieson et al. (2012) ). For online learning, Flaxman et al. (2005) introduce online convex optimization with "bandit" feedback. It considers that, in each round, the learner first plays a point in a convex set and then it suffers a loss which is the function value at the point that the learner plays. The learner only knows the function value at the point it plays. The goal of the learner is to minimize a quantity called "regret". Flaxman et al. (2005) make the minimal assumption that the loss function in each round is convex while allow the loss functions be adversarially different in different rounds. Many follow-up papers consider different variants of the setting (see e.g., Agarwal et al. (2010); Shamir (2013; 2017)). Zeroth order optimization also has a connection with some biological evolution (see Rechenberg (1989) and Section 2 of Salimans et al. (2017)). As the result, there are many zeroth order algorithms named as evolution strategies (e.g., Glasmachers et al. (2010); Hansen & Ostermeier (2001); Wierstra et al. (2014)). We also notice that many evolution strategies consider sampling from full covariance Gaussian distributions.

Substantial progress has been made in zeroth order optimization. Ghadimi & Lan (2013); Lian et al. (2016); Liu et al. (2018b); Ji et al. (2019) propose some zeroth order algorithms for non-convex optimization. On the other hand, there are some efforts in integrating the "gradient-free" technique into some first-order algorithms. For example, there are a zeroth order counterpart of Alternative Direction method of Multipliers (Liu et al. (2018a)), a zeroth order version of the Frank-Wolfe method (Balasubramanian & Ghadimi (2018)), and a zeroth order version of Adam (Chen et al. (2019)). The design principle is basically replacing the step of computing a gradient with that of constructing the pseudo one. Another direction is by reducing or removing the dependency of convergence rate on the dimension. Since without further assumptions, the convergence rate of any zeroth order optimization algorithms depends on the dimension (see e.g., Nesterov & Spokoiny (2017); Duchi et al. (2012; 2015); Shamir (2013; 2017)), a zeroth order algorithm can suffer from high dimensionality. To deal with the issue, Wang et al. (2018) assume a statistical model for an optimization problem and exploit the assumption to design some algorithms whose convergence rates do not depend on the dimension. Other directions in zeroth order optimization include designing a new estimator of the pseudo-gradient (Choromanski et al. (2019b); Rowland et al. (2018)), which can lead to a smaller variance of the estimator compared to the standard Monte-Carlo approach.

In this paper, we consider combining different ways of sampling perturbed vectors, which to our knowledge is a new direction. As mentioned in the introduction, some recent works propose constructing pseudo gradients by sampling from a non-isotropic Gaussian distributions whose covariance reflects a local geometry around the current update (e.g. Maheswaranathan et al. (2019); Choromanski et al. (2019a); Ye et al. (2019)). Though the methods are interesting and well motivating, it is unclear if they can guarantee the convergence. We tackle the issue by proposing sampling from a mixture of distributions. Our method builds on a technique in bandit literature called EXP3.P (Auer et al. (2002)), which is used for adaptively selecting a sampling scheme during the optimization process.

## 3 PRELIMINARIES

**Notation:** For a vector $\boldsymbol{u} \in \mathbb{R}^d$, we denote $\boldsymbol{u}^2$ as the element-wise square and denote $\text{diag}(\boldsymbol{u})$ as the $d \times d$ diagonal matrix whose diagonal elements are the elements of $\boldsymbol{u}$. We use $u_m$ to denotes its $m_{th}$ element of $\boldsymbol{u}$, which should be distinguished from $\boldsymbol{u}_t$ that stands for the vector $\boldsymbol{u}$ in iteration $t$.

Many zeroth order optimization algorithms construct a "pseudo" gradient in the following ways. First, the so called *Gaussian smoothing* is applied to the underlying function $f(\cdot)$ to get a new function $f_\sigma(\cdot)$,

$$f_\sigma(\boldsymbol{w}) := \mathbb{E}_{\boldsymbol{v} \sim \mathcal{N}(0, \boldsymbol{I}_d)}[f(\boldsymbol{w} + \sigma \boldsymbol{v})]. \tag{2}$$

The new function $f_\sigma(\cdot)$ is differentiable and the gradient of $f_\sigma(\cdot)$ can be shown to be (see e.g., Nesterov & Spokoiny (2017))

$$\begin{aligned}
\nabla f_\sigma(\boldsymbol{w}) &= \frac{1}{\sigma} \mathbb{E}_{\boldsymbol{v} \sim \mathcal{N}(0, \boldsymbol{I}_d)} \big[ f(\boldsymbol{w} + \sigma \boldsymbol{v}) \boldsymbol{v} \big] \\
&= \frac{1}{\sigma} \mathbb{E}_{\boldsymbol{v} \sim \mathcal{N}(0, \boldsymbol{I}_d)} \big[ f(\boldsymbol{w} + \sigma \boldsymbol{v}) \boldsymbol{v} - f(\boldsymbol{w}) \boldsymbol{v} \big],
\end{aligned} \tag{3}$$

where the second equality is due to that $\boldsymbol{v} \in \mathbb{R}^d$ is a zero mean vector. The pseudo gradient $\nabla f_\sigma(\boldsymbol{w})$ can be viewed as the expectation of the perturbed vectors weighted by the corresponding function values at the near-by points of $\boldsymbol{w}$. Since it involves the expectation, in practice one uses a mini-batch of perturbed vectors to estimate $\nabla f_\sigma(\boldsymbol{w})$ (like Algorithm 1 does). The parameter $\sigma$ controls a trade-off. The larger $\sigma$, the larger bias of $f_\sigma(\cdot)$ (i.e., larger $|f_\sigma(\cdot) - f(\cdot)|$), but large $\sigma$ might help faster convergence (to the optimum of $f_\sigma(\cdot)$) (Nesterov & Spokoiny (2017); Duchi et al. (2012)). Since the algorithms actually optimize $f_\sigma(\cdot)$ instead of the original function $f(\cdot)$, one should not choose too large $\sigma$, as the difference between $f_\sigma(\boldsymbol{w})$ and $f(\boldsymbol{w})$ increases as $\sigma$ increases.

Several algorithms in the literature can be written as a generic scheme equipped with different kinds of perturbation. Specifically, many existing algorithms can be generated from Algorithm 2 by using different sampling oracles. For example, we can see that Algorithm 1 is an instance of Algorithm 2 with sampling oracle on step 5 being Algorithm 3. Moreover, the recently proposed methods Maheswaranathan et al. (2019); Choromanski et al. (2019a); Ye et al. (2019) can all be viewed as some

instances of Algorithm 2. As an illustration, an algorithm in Ye et al. (2019) is an instance of Algorithm 2 with the sampling oracle on step 5 being Algorithm 4. Ye et al. (2019) argue that the covariance matrix $\boldsymbol{\Sigma}$ should be the inverse of the Hessian. However, only the function value oracle is available in zeroth order optimization, let alone the second-order information is provided. Ye et al. (2019) propose to use the second moment quantity of Adam optimization algorithm (Kingma & Ba (2015)) to construct a surrogate of the Hessian. However, when the perturbed vectors are sampled from a non-isotropic Gaussian distribution, it may not lead to the convergence to an optimal point even when the function is convex.

---

**Algorithm 2** A template of zeroth order algorithms.

---

1: **Require:** a sampling oracle.
2: **Require:** hyper-parameters $\sigma$, $K$, and $\eta$.
3: **Init:** $\boldsymbol{w}_0 \in \mathbb{R}^d$.
4: **for** $t = 0$ to $T$ **do**
5:      Update $\boldsymbol{\Sigma}_t \leftarrow$ SAMPLINGORACLE.
6:      Sample $K$ perturbed vectors $\boldsymbol{v}_t^k \sim \mathcal{N}(0, \boldsymbol{\Sigma}_t)$ for each $k \in [K]$.
7:      Construct pseudo-gradient $\boldsymbol{g}_t := \frac{1}{K\sigma} \sum_{k=1}^{K} \left( f(\boldsymbol{w}_t + \sigma \boldsymbol{v}_t^k) - f(\boldsymbol{w}_t) \right) \boldsymbol{v}_t^k$.
8:      Update parameter $\boldsymbol{w}_{t+1} = \boldsymbol{w}_t - \eta \boldsymbol{g}_t$.
9: **end for**

---

---

**Algorithm 3** SAMPLINGORACLE: ISOTROPIC COVARIANCE

---

1: **Output:** $\boldsymbol{\Sigma}_t = \boldsymbol{I}_d$.

---

---

**Algorithm 4** SAMPLINGORACLE: ADAM-STYLE HESSIAN Ye et al. (2019).

---

1: **Require:** hyper-parameter $\alpha$ and $\epsilon$.
2: **Input:** pseudo gradient $\boldsymbol{g}_t$.
3: **Init:** $\boldsymbol{D}_0 = \epsilon \boldsymbol{I}_d$.
4: $\boldsymbol{D}_{t+1} = \alpha \boldsymbol{D}_t + (1 - \alpha)\mathrm{diag}(\boldsymbol{g}_t)^2$.
5: **Output:** $\boldsymbol{\Sigma}_t = $ inverse $(\boldsymbol{D}_t)$.

---

## 4 PROPOSED METHOD

Consider that there are $M$ different sampling oracles available. Each oracle (indexed by $m \in [M]$) maintains its covariance matrix $\boldsymbol{\Sigma}^m$ and one of them always outputs the isotropic covariance (e.g., Algorithm 3). So each oracle is actually associated with a Gaussian distribution which the perturbed vectors can be sampled from.

We propose Algorithm 5. The idea is sampling the perturbed vectors $\boldsymbol{v}_t^k$ from a mixture of Gaussian distributions which are individually updated by their corresponding sampling oracles,

$$\boldsymbol{v}_t^k \sim \sum_{m=1}^{M} p_m \mathcal{N}(0, \sigma \boldsymbol{\Sigma}_t^m) \in \mathbb{R}^d, \; \forall k \in [K] \tag{4}$$

where we should have the weight of each mixture component $p_m$ satisfy

- $p_m \in [0, 1]$ and $\sum_{m=1}^{M} p_m = 1$.
- $\boldsymbol{p} \in [0, 1]^M$ adapts over time.

To sample $\boldsymbol{v}_t^k$ from the mixture, Algorithm 5 first samples an index $m_t \in [M]$ with each $m \in [M]$ following the categorical distribution indicated by $\boldsymbol{p}$. After that, it samples $\boldsymbol{v}_t^k$ from the Gaussian distribution indexed by $m_t$. The pseudo gradient is then constructed and is used to obtain the next update $\boldsymbol{w}_{t+1}$. At the end of an iteration, the covariance of the Gaussian distribution in which the vectors are sampled from is updated before continuing the next iteration. Notice that only one perturbation scheme (one sampling oracle) is used in each iteration of Algorithm 5; the computational complexity is basically the same as that of an algorithm which always uses a fixed sampling oracle (except the overhead for possibly updating the mixture weight $\boldsymbol{p}$).

---

**Algorithm 5** A mixture of evolution strategies.

---

1: **Require:** $M$ sampling oracles with one oracle being Algorithm 3.
2: **Require:** hyper-parameters $\sigma$, $K$, and $\eta_t$.
3: **Init:** $\boldsymbol{w}_0 \in \mathbb{R}^d$ and a vector $\boldsymbol{p} \in \Delta_M$ ($\Delta_M$ stands for the $M$-dimensional simplex).
4: **for** $t = 0$ to $T - 1$ **do**
5:     Select an index $m_t \in [M]$ with each $m \in [M]$ following the categorical distribution $m \sim p_m$.
6:     Sample $K$ perturbed vectors $\boldsymbol{v}_t^k \sim \mathcal{N}(0, \boldsymbol{\Sigma}_t^{m_t})$ for each $k \in [K]$.
7:     Construct pseudo-gradient $\boldsymbol{g}_t := \frac{1}{K\sigma} \sum_{k=1}^{K} \left( f(\boldsymbol{w}_t + \sigma \boldsymbol{v}_t^k) - f(\boldsymbol{w}_t) \right) \boldsymbol{v}_t^k$.
8:     Update parameter $\boldsymbol{w}_{t+1} = \boldsymbol{w}_t - \eta_t \boldsymbol{g}_t$.
9:     Update $\boldsymbol{\Sigma}_{t+1}^{m_t} \leftarrow \textsc{SamplingOracle } m_t$.
10: **end for**

---

**Algorithm 6** EXP3.P algorithm (Auer et al. (2002)).

---

1: **Require:** hyper-parameters $\eta_{exp3}, \nu, \gamma$.
2: **Init:** A vector $\boldsymbol{p} \in \Delta_M$ ($\Delta_M$ stands for the $M$-dimensional simplex).
3: **for** $t = 0$ to $T - 1$ **do**
4:     Select an index $m_t \in [M]$ with each $m \in [M]$ following the categorical distribution $m \sim p_m$.
5:     Observe the loss $\ell_t^{(m_t)}$ for picking $m_t$.
6:     Compute the estimated loss for each $m$ as $\tilde{\ell}_t^{(m)} := \frac{\ell_t^{(m)} \mathbb{1}[m_t = m] + \nu}{p_m}$ and update the estimated cumulative loss $L_t^{(m)} := \sum_{s=0}^{t} \tilde{\ell}_s^{(m)}$.
7:     Update the weight of each component $p_m = (1 - \gamma) \frac{\exp(-\eta_{exp3} L_t^{(m)})}{\sum_{m=1}^{M} \exp(-\eta_{exp3} L_t^{(m)})} + \frac{\gamma}{M}$.
8: **end for**

---

Natural questions regarding to the algorithm are (1) which oracle should be called for sampling the perturbed vectors $\boldsymbol{v}_t^k$ in each iteration, and (2) how to update $\boldsymbol{p}$ over time. Intuitively, one should add more weight to the perturbation scheme that works well and decrease the weight of other perturbation schemes. A concern is that if one chooses a perturbation scheme for the update in a round, then one would not know the outcomes (i.e., the progress of optimization) of choosing other schemes in that round. Therefore, we propose to use EXP3.P algorithm in Auer et al. (2002) (see Algorithm 6) for adaptively selecting a perturbation scheme (i.e., step 5 of Algorithm 5) and updating the mixture weight $\boldsymbol{p}$ accordingly. To use EXP3.P algorithm, we define the loss of choosing $m_t \in [M]$ as $\ell_t^{(m_t)} := f(\boldsymbol{w}_{t+1}) - f(\boldsymbol{w}_t) + \eta_t LG$, where $L$ and $G$ are constants defined in the later subsection. One can see that it is small when the function value at the next update $\boldsymbol{w}_{t+1}$ due to choosing $m_t$ is smaller than $f(\boldsymbol{w}_t)$ (which suggests a progress). On the other hand, the loss is large when choosing $m_t$ does not lead to a decrease of the function value. As the next update $\boldsymbol{w}_{t+1}$ depends on the choice of the sampling oracle $m_t$, in the following we also denote $\boldsymbol{w}_{t+1}^{(m_t)}$ to explicitly state $\boldsymbol{w}_{t+1}$'s dependency on the selected sampling oracle $m_t$. One can view that EXP3.P runs synchronously with Algorithm 5 in the way that step 4 of EXP3.P implements step 5 of Algorithm 5 and that step 5 and the subsequent steps of EXP3.P are conducted after step 8 of Algorithm 5 is finished (so that EXP3.P can get $\ell_t^{(m_t)}$ for updating its mixture weight $\boldsymbol{p}$).

## 4.1 Intuition in the design of the proposed algorithm

In this section, we explain the idea behind the design of the proposed algorithm. Let us begin by introducing some notations. We denote the $M$ dimensional loss vector $\ell_t := [\ell_t^{(1)}, \ell_t^{(2)}, \dots, \ell_t^{(M)}]$, in which each element $\ell_t^{(m)}$ represents the loss in iteration $t$ if the algorithm chooses to sample the perturbed vectors from the oracle $m$. So $\ell_t^{(m_t)}$ stands for the loss of EXP3.P for choosing $m_t$. We will make the assumptions that the function $f(\cdot)$ is non-negative, $L$-Lipschitz convex and the pseudo gradient $g_t$ satisfies $\|g_t\| \le G, \forall t$ for a constant $G$.

Our proposed method requires a distribution in the mixture to be an isotropic Gaussian distribution. So in the following, w.l.o.g. we assume that $m = 1$ refers to the corresponding oracle. The following is a convergence guarantee regarding to the algorithm that always samples the perturbed vectors from the distribution.

**Theorem 1.** *(Theorem 6 of Nesterov & Spokoiny (2017)) Algorithm 1 guarantees $\mathbb{E}[f(\hat{\boldsymbol{w}}_T)] - f^* \leq \epsilon$ when $T = \frac{4(d+4)^2 L^2 R^2}{\epsilon^2}$, $\sigma \leq \frac{\epsilon}{2Ld^{1/2}}$, $K = 1$, and $\eta_t = \frac{R}{(d+4)(T+1)^{1/2}L}$, where $\hat{\boldsymbol{w}}_T := \arg\min_{\boldsymbol{w}}[f(\boldsymbol{w}) : \boldsymbol{w} \in \{\boldsymbol{w}_0, \boldsymbol{w}_2, \ldots, \boldsymbol{w}_T\}]$, $R$ being the bound of $\|\boldsymbol{w}_0 - \boldsymbol{w}^*\| \leq R$, and $\boldsymbol{w}^*$ represents one of the optimal solutions.*

Now let us replicate the theoretical statement of EXP3.P.

**Theorem 2.** *(Theorem 3.3 in Sebastien Bubeck (2012);see also Auer et al. (2002)) Assume that each element of the loss vector $\ell_t$ is in $[0, c']$. If the parameters of EXP3.P are chosen so that $\nu = \sqrt{\frac{\log(M\delta^{-1})}{TM}}$, $\eta_{exp3} = 0.95\sqrt{\frac{\log M}{TM}}$, and $\gamma = 1.05\sqrt{\frac{M\log M}{T}}$, then with probability at least $1 - \delta$, $Regret_T^{(m)} := \sum_{t=1}^T \ell_t^{(m_t)} - \sum_{t=1}^T \ell_t^{(m)} \leq c\sqrt{TM\log(M\delta^{-1})}$, for any fixed $m \in [M]$, where the constant $c$ is independent from $T, M, \delta$.*

Notice that the theorem guarantees a strong result; the regret bound holds for any fixed benchmark $m \in [M]$. To use the theorem, let us verify the non-negativeness of the $\ell_t$. By the Lipschitz assumption of the algorithm, the loss in any round $t$ is guaranteed to be in the range of $[0, c']$ with $c' = 2\eta_t LG$. This is because $\ell_t^{(m_t)} := f(\boldsymbol{w}_{t+1}^{(m_t)}) - f(\boldsymbol{w}_t) + \eta_t LG \geq -L\|\boldsymbol{w}_{t+1}^{(m_t)} - \boldsymbol{w}_t\| + \eta_t LG \geq 0$, where the first inequality is by the $L$-Lipschitzness of the function $f(\cdot)$ and the second one is by the update rule and the assumption that $g_t$ satisfies $\|g_t\| \leq G$. Similarly, one can also show that $\ell_t \leq 2\eta_t LG$. So one can apply the theoretical guarantee of EXP3.P.

Let us denote the best sampling oracle as $m^* \in [M]$, which is the best choice if it commits to a fixed perturbation scheme during optimization. That is, $m^*$ achieves the smallest optimality gap, $f(\boldsymbol{w}_t^{(m^*)}) - \min_{\boldsymbol{w}} f(\boldsymbol{w})$, after a sufficiently large $t \geq \bar{T}$ iterations, among all the fixed perturbation schemes. Let us also denote $T_1 \geq \bar{T}$ as the number of iterations for the algorithm which always chooses $m = 1$ (the isotropic Gaussian) to achieve the optimal gap being less than $\epsilon$. Now observe that the average regret, $\frac{Regret_T^{(m^*)}}{T}$, approaches 0 as the number of iterations $T \to \infty$. So it implies that after sufficient number of iterations, say $T_0$, the majority of the choice should converge to the optimal $m^*$ with a high probability. By conducting the proposed algorithm with number of iterations $T = \max(c_* T_1, T_0)$ with some $c_* \geq 1$, it will make the proposed algorithm eventually behaves as a convergent algorithm which uses a fixed sampling oracle. Therefore, one might show that the proposed algorithm converges with a high probability.

## 4.2 TWO AUGMENTATIONS

In this section, we propose two augmentations of the proposed method. The first augmentation is about changing the definition of the loss $\ell_t^{(m_t)}$. We now allow the loss to be negative and consider defining $\ell_t^{(m_t)} := \frac{f(\boldsymbol{w}_{t+1}^{(m_t)}) - f(\boldsymbol{w}_t)}{f(\boldsymbol{w}_t)}$ instead. The change leads to a more aggressive update of the mixture weight. Specifically, if the function value at the next update $\boldsymbol{w}_{t+1}^{(m_t)}$ is smaller than $f(\boldsymbol{w}_t)$, then $\ell_t^{(m_t)}$ is negative (assume that $f(\cdot)$ is non-negative) and consequently the weight $p_{m_t}$ is much more likely to increase or increases more using the new loss.

For the other augmentation, it is due to an observation that the variance of the loss $\ell_t^{(m_t)}$ can be large, since the algorithm only uses $K$ samples to estimate the pseudo gradient $\nabla f_\sigma(\cdot)$ (recall that exactly computing $\nabla f_\sigma(\cdot)$ involves expectation). As a result, it might be the case that the selected sampling oracle $m_t$ is actually the right choice for decreasing the function value if $\nabla f_\sigma(\cdot)$ can be computed exactly, but somehow the realization of $\ell_t^{(m_t)}$ is large due to the high variance and consequently the weight of the oracle $p_{m_t}$ is severely penalized instead. To deal with this issue, we consider only calling EXP3.P algorithm for selecting a sampling oracle every $\tau$ iterations. This means that a sampling oracle is chosen for a consecutive $\tau$ iterations before the possible switch. Combining the first augmentation, we define the loss $\ell_t^{(m_{t-\tau})}$ as $\ell_t^{(m_{t-\tau})} := \frac{f(\boldsymbol{w}_t^{(m_{t-\tau})}) - f(\boldsymbol{w}_{t-\tau})}{f(\boldsymbol{w}_{t-\tau})}$ for $(t \mod \tau) == 0$. Algorithm 7 (MIXTURE) in Appendix A shows the proposed algorithm equipped with the two augmentations.

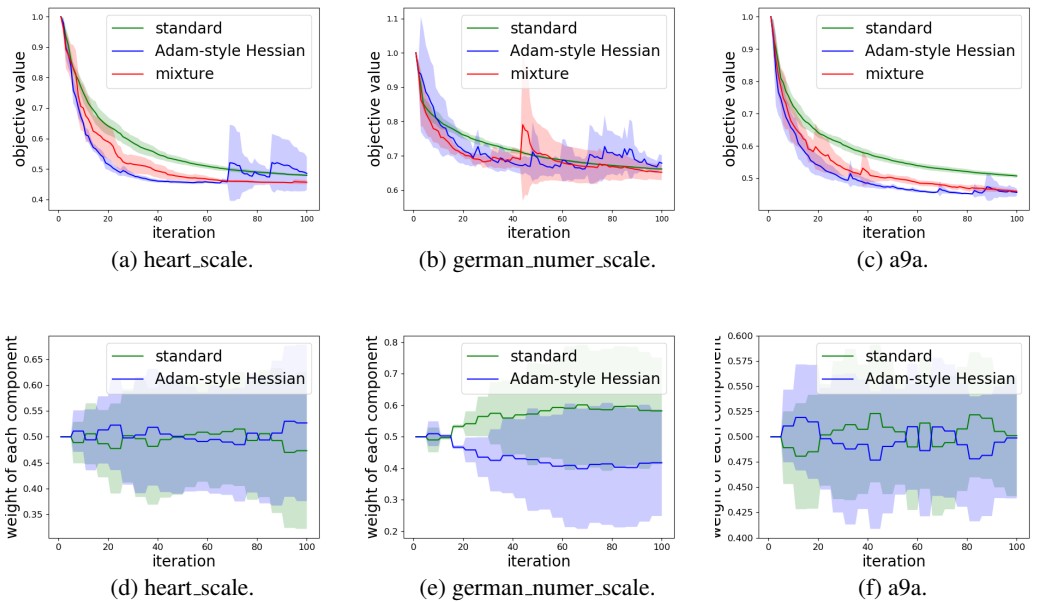

(a) heart_scale.  (b) german_numer_scale.  (c) a9a.

(d) heart_scale.  (e) german_numer_scale.  (f) a9a.

Figure 1: Convex optimization $\min_w f(w) := \min_w \frac{1}{n} \sum_{i=1}^{n} \max \left( 0, 1 - y_i(w^\top x_i) \right)^2$.

## 5 EXPERIMENTS

We compare our algorithm (MIXTURE) with two baselines: (1) Algorithm 2 with line 5 implemented by Algorithm 3 (STANDARD), and (2) Algorithm 2 with line 5 implemented by Algorithm 4 (ADAM-STYLE HESSIAN). The mixture of our algorithm consists of two Gaussian distributions which are updated by Algorithm 3 and Algorithm 4 respectively. So if it always chooses a specific oracle, then it would be equivalent to one of the two baselines.

**Convex optimization:** We consider an empirical risk minimization with squared of hinge loss $f(w) := \frac{1}{n} \sum_{i=1}^{n} \max \left( 0, 1 - y_i(w^\top x_i) \right)^2$, where each $(x_i, y_i)$ is a feature/label pair. We compare the algorithms on heart_scale, german.numer_scale, and a9a datasets [1], which are all with binary labels $y_i = \{\pm 1\}$. All the algorithms can only access to the oracle of function value evaluations. We use the same set of hyper-parameters for different datasets and repeated runs in the experiments. They are

- (Same for all the algorithms) step size $\eta = 0.01$, number of perturbed vectors $K = 4$, and perturbation parameter $\sigma = 0.001$.
- (ADAM-STYLE HESSIAN) mixing parameter $\alpha = 0.9$ and $\epsilon = 1.0$.
- (Algorithm 7) duration $\tau = 5$, step size $\eta_{exp3} = 0.1$, $\gamma = 0$, and $\nu = 0$.

For each dataset, we repeat the experiment 5 times and report the average and the standard deviation. Figure 1 shows the results. The top row on Figure 1 plots the objective value versus iteration while the bottom row plots the weight of each component in the mixture throughout iterations. We see that our algorithm MIXTURE is competitive to any of its mixture components. On heart_scale dataset, ADAM-STYLE HESSIAN makes a larger improvement in the early stage, but it suffers from a fluctuation when the update is close to an optimum, while the proposed MIXTURE is stable and converges to an optimum. On german.numer_scale, MIXTURE and ADAM-STYLE HESSIAN decreases function values more than STANDARD does in the first few iterations, but have some fluctuations of the function values later. Yet, an interesting observation is that the weight of component STANDARD in

---

[1]All are available online https://www.csie.ntu.edu.tw/~cjlin/libsvmtools/datasets/.

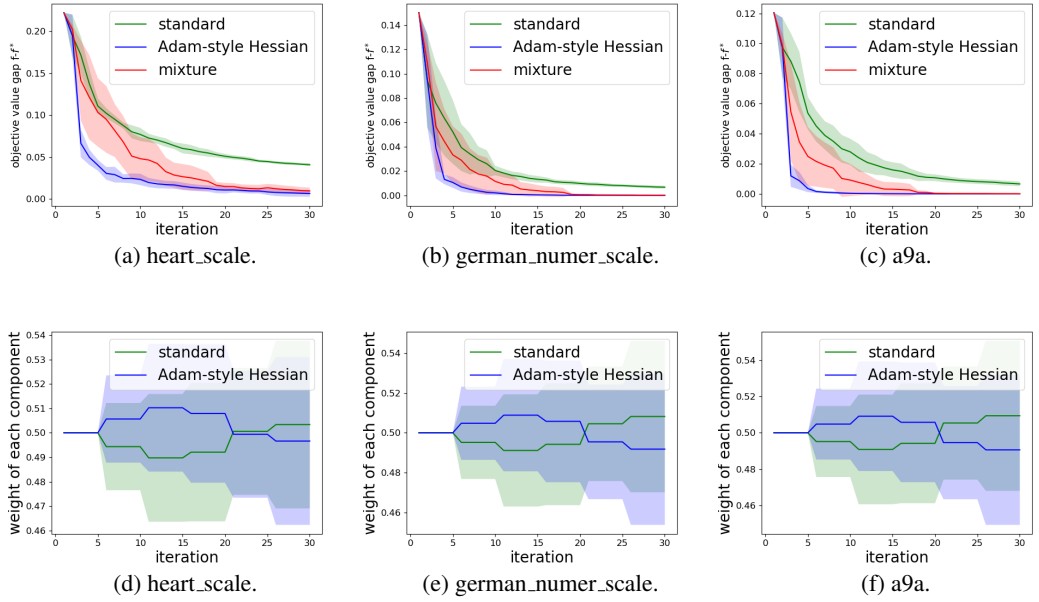

Figure 2: Non-convex optimization $\min_w f(w) := \min_w \frac{1}{n} \sum_{i=1}^{n} \frac{1}{1+\exp(y_i w^\top x_i)}$.

MIXTURE increases in the later iterations, which suggests that MIXTURE might rely on the oracle STANDARD to reach to an $\epsilon$-optimal point in this case, as solely using ADAM-STYLE HESSIAN may not be guaranteed the convergence. The observation also suggests that the proposed algorithm can adaptively choose a sampling oracle in an effective way.

**Non-convex optimization:** We also consider a non-convex optimization problem with sigmoid loss function $f(w) := \frac{1}{n} \sum_{i=1}^{n} \frac{1}{1+\exp(y_i w^\top x_i)}$, which was also considered in Daneshmand et al. (2018). We use the same set of datasets as the convex optimization experiments but we relabel those data with $y_i = -1$ as $y_i = 0$ (namely $y_i = \{0, 1\}$ here). The same hyper-parameter values are used, except that we set $\eta = 1$ for all the algorithms in this part of the experiments. Figure 2 shows the results. We see that always sampling from the inverse of ADAM-STYLE HESSIAN performs the best. On the other hand, MIXTURE learns a larger weight of ADAM-STYLE HESSIAN (compared to that of the sampling oracle STANDARD) in the first few iterations, which implies that it identifies the component that works best in the early stage of optimization and consequently converges faster than STANDARD.

# 6 CONCLUSION

In this work, we propose sampling from a mixture of distributions for zeroth order optimization. The propose method is modular and is complementary to existing works. It allows any proposals of sampling oracle and is an easy-to-use "meta" algorithm that provides a reliable way to combine different perturbation schemes and different heuristics. We believe that the proposed method helps in the applications of hyper-parameter search, reinforcement learning, or black-box adversarial attacks where zeroth order optimization has been used as a building block or as a sub-module of the algorithms. An interesting future work is designing a life-long zeroth order optimization algorithm. Consider the setting that a series of similar optimization problems needed to be solved one by one over time. Is it possible to extend the idea of sampling from a mixture to the life-long optimization setup by assuming that the best mixture weight $p$ for each optimization problem is similar?

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

## A   AUGMENTED MIXTURE OF EVOLUTION STRATEGIES.

---

**Algorithm 7** MIXTURE

---

1:  **Require:** $M$ sampling oracles with one oracle being Algorithm 3.
2:  **Require:** hyper-parameters $\sigma$, $K$, $\eta$, $\eta_{exp3}$, $\nu$, $\gamma$ and $\tau$.
3:  **Init:** $\boldsymbol{w}_0 \in \mathbb{R}^d$, $\boldsymbol{p} = [\frac{1}{M}, \frac{1}{M}, \ldots, \frac{1}{M}]$, $m = 1$.
4:  **for** $t = 0$ to $T$ **do**
5:      **if** $(t \bmod \tau) == 0$ **then**
6:          Select an index $m_t \in [M]$ with each $m \in [M] \sim p_m$.
7:          **if** $t > 0$ **then**
8:              $\ell_t^{(m_{t-\tau})} := \frac{f(\boldsymbol{w}_t^{(m_{t-\tau})}) - f(\boldsymbol{w}_{t-\tau})}{f(\boldsymbol{w}_{t-\tau})}$.
9:              Compute the estimated loss for each $m$ as $\tilde{\ell}_t^{(m)} := \frac{\ell_t^{(m)} \mathbb{1}[m_{t-\tau}=m]+\nu}{p_m}$ and update the estimated cumulative loss $L_t^{(m)} := \sum_{s=0}^{t} \tilde{\ell}_s^{(m)}$.
10:              Update the weight of each component $p_m = (1 - \gamma)\frac{\exp(-\eta_{exp3}L_t^{(m)})}{\sum_{m=1}^{M} \exp(-\eta_{exp3}L_t^{(m)})} + \frac{\gamma}{M}$.
11:          **end if**
12:          $m \leftarrow m_t$.
13:      **end if**
14:      Sample $K$ perturbed vectors $\boldsymbol{v}_t^k \sim \mathcal{N}(0, \boldsymbol{\Sigma}_t^m)$ for each $k \in [K]$.
15:      Construct pseudo-gradient $\boldsymbol{g}_t := \frac{1}{K\sigma} \sum_{k=1}^{K} \left(f(\boldsymbol{w}_t + \sigma\boldsymbol{v}_t^k) - f(\boldsymbol{w}_t)\right)\boldsymbol{v}_t^k$.
16:      Update parameter $\boldsymbol{w}_{t+1} = \boldsymbol{w}_t - \eta\boldsymbol{g}_t$.
17:      Update $\boldsymbol{\Sigma}_{t+1}^m \leftarrow$ SAMPLINGORACLE $m$.
18:  **end for**

---

