# OpenReview forum: "Zeroth Order Optimization by a Mixture of Evolution Strategies"
_ICLR.cc/2020/Conference — Reject_

### Official Review · AnonReviewer3 · 2019-10-21
**Official Blind Review #3**

**Rating:** 3

**Review:**

The submission is proposing an adaptive derivative-free optimization method. In this method, the sampling covariances are adapted between different covariance adaptation heuristics. The main intuition is seeing each algorithm as an arm in multi-armed bandit (MAB) setting. Moreover, authors use EXP3.P as an online learning mechanism.

The idea and intuition is definitely interesting. Although the idea of adapting DFO using MAB has been already explored, using MAB for choosing covariance adaptation scheme is definitely novel and interesting. However, the submission has various issues which need to be fixed.

The claim that the existing adaptation mechanisms have no guarantees is not correct. (Ye et al, 2019) provides convergence guarantees. Moreover, most of the existing adaptive optimization method convergence guarantees can be extended to DFO in a straightforward manner.

The main claim ("we can show that our method converges with a high probability when the underlying function is convex") is not completely substantiated. The high-probability no-regret guarantee (Theorem 2) the authors state is using the iterates (w_t) as an input. It shows that given adversarially chosen iterates (w_t), the Exp3.p has no regret. However, this says nothing about the convergence of these iterates (w_1,...,w_t). I think it is not hard to prove the main claim; however, the reasoning provided in the paper is incomplete. More importantly, both Thm.1 and Thm.2 are specifying the sample complexity of the methods. Whereas, authors only give rather asymptotic argument. Authors should 1) rigorously prove the convergence guarantee as an additional theorem. And, 2) give sample complexity of the proposed method.

Another issue is the empirical validation. It is clear from the results that the adam-style Hessian method outperforms the proposed method. This is counter-intuitive and no explanation (theoretical or empirical) is given for this discrepancy.

As a minor question: Why do you only update one method after each gradient update in Line 9 of Alg5? It is an unbiased estimate of the gradient, hence all sampling oracles can update itself using this estimate?

**Experience Assessment:**

I have published one or two papers in this area.

**Review Assessment: Checking Correctness Of Derivations And Theory:**

I carefully checked the derivations and theory.

**Review Assessment: Checking Correctness Of Experiments:**

I assessed the sensibility of the experiments.

**Review Assessment: Thoroughness In Paper Reading:**

I read the paper thoroughly.

---

### Official Review · AnonReviewer1 · 2019-10-22
**Official Blind Review #1**

**Rating:** 1

**Review:**

This paper proposes to change the sampling scheme of evolutionary strategies to sample from a mixture of distributions (instead of the standard way of sampling from one Gaussian).
Although this seems to provide slightly better performance, the contribution is incremental and more importantly the paper lacks clarity, especially regarding the theory part. The experiments are also not on par with what’s typically expected in the literature. Overall, this paper is not ready for acceptance at a venue such as ICLR.

EXP3.P algorithm
This is one of the key components of the algorithm presented by the authors but it is not properly explained in the paper. In fact, the algorithm implemented by the authors is very unclear to me. Below I quote the description given by the authors:
“One can view that EXP3.P runs synchronously with Algorithm 5 in the way that step 4 of EXP3.P implements step 5 of Algorithm 5 and that step 5 and the subsequent steps of EXP3.P are conducted after step 8 of Algorithm 5 is finished“
This is a very poor description of the algorithm. The authors should give a clear pseudo-code to explain the algorithm.

Theoretical guarantees
1) Again, the paper is very unclear on what algorithm is being implemented. The authors claim they can directly use the guarantees provided by Bubeck et al but this is unclear to me given that the algorithm is never fully stated. This needs to be clarified.
2) The authors claim asymptotic convergence, specifically they claim “Therefore, one might show that the proposed algorithm converges with a high probability.“. This proof should be provided in the paper, not left as an exercise to the reader...
3) Does Theorem 2 (Theorem 3.3 from Bubeck) require convexity of the function? Can you comment on guarantees for non-convex functions?

Prior work
The authors do not give a clear description of prior work. There is a large literature on proving convergence guarantees under weaker assumptions than discussed in the paper, see e.g.
Y. Diouane, S. Gratton, and L. N. Vicente. Globally convergent evolution strategies. Math.
Program., 152:467–490, 2015.
Y. Diouane, S. Gratton, and L. N. Vicente. Globally convergent evolution strategies for
constrained optimization. Comput. Optim. Appl., 62:323–346, 2015.
Vincente. Worst case complexity of direct search

Experiments
This is also a weak part of the paper.
1) The authors only run experiments on very small datasets. Regarding the non-convex task, it would be much more interesting to consider a reinforcement learning task where evolutionary methods typically perform well.
2) You should include stronger baselines too, e.g. https://arxiv.org/abs/1806.10230.
3) An interesting question to look at would be to analyze the behavior of the algorithm as a function of the number of mixture components. Also, what is the computational increase in terms of the number of components? Consider giving plots in terms of computation time.


**Experience Assessment:**

I have read many papers in this area.

**Review Assessment: Checking Correctness Of Derivations And Theory:**

I carefully checked the derivations and theory.

**Review Assessment: Checking Correctness Of Experiments:**

I assessed the sensibility of the experiments.

**Review Assessment: Thoroughness In Paper Reading:**

I read the paper thoroughly.

---

### Official Review · AnonReviewer2 · 2019-10-23
**Official Blind Review #2**

**Rating:** 1

**Review:**

This paper is motivated to improve the performance of zeroth order stochastic search by incorporating a gaussian mixture as the candidate solution sampler. It is (almost) not possible to improve the state of the art zeroth order optimizer in all aspects, hence I expect a description in which situation the authors try to improve the performance by the proposed technique. However, it is not mentioned in the paper.

The experiments are far away from convincing. The authors pick only two very simple functions, convex one and non-convex but very simple one. The experimental results are compared only with the two baselines of the proposed algorithm. I have to say that the experimental design of this paper must be reconsidered to conclude and derive the goodness of the proposed algorithm.

First of all, the effect of introducing non-identity covariance matrix is usually to tackle the ill-conditioning of the objective function, as is well discussed in papers addressing the covariance matrix adaptation in evolution strategies (CMA-ES) by Hansen and his co-authors. However, the objective function doesn't seem to be so. Therefore, this effect can not be tested from this experiments. It is not at all clear what the authors want to claim from these experiments.

If the non-isotropic part is not important but the scaling factor of the covariance matrix matters (for the step-size adaptation effect), one must compare algorithms using isotropic covariance matrix but with step-size adaptation mechanisms. For example, random pursuit (Stich, SIOPT 2013) is a hill-climbing algorithm with randomly sampled direction with a line search. It also has a convergence results on convex functions. (1+1)-evolution strategy is another random direction hill-climbing algorithm with a step-size control mechanism. (Morinaga and Akimoto, FOGA 2019) has shown its linear convergence of this algorithm on strongly convex functions and beyond. These algorithms works reasonably well without tuning the hyper-parameters. In the proposed approach, there are several hyper-parameters that needs to be tuned to guarantee the convergence and perform reasonably. What is then the goodness of the proposed algorithm compared with above mentioned algorithms?

Theorems provided in the paper doesn't show the goodness of the proposed algorithm, but it tells that the idea of having a mixture model is not a bad idea. I do not see from these arguments the reason why a mixture model leads to a better performance than the baseline. Nonetheless, the theoretical analysis of this work is not rigorous, and if I understand correctly, the convergence (and its rate) of the algorithm is not formally proved.

**Experience Assessment:**

I have published in this field for several years.

**Review Assessment: Checking Correctness Of Derivations And Theory:**

I assessed the sensibility of the derivations and theory.

**Review Assessment: Checking Correctness Of Experiments:**

I carefully checked the experiments.

**Review Assessment: Thoroughness In Paper Reading:**

I read the paper at least twice and used my best judgement in assessing the paper.

---

### Decision · Program_Chairs · 2019-12-19

**Decision:**

Reject

**Comment:**

The paper proposes an adaptive sampling mechanism for zeroth order optimization that samples perturbed points from a mixture distribution with asymptotic convergence guarantees. The reviewers raised issues regarding the clarity of presentation, potential problems with the proofs, and simplicity of the experimental setup. The authors did not provide a response. Overall, the reviewers agree that the quality of the paper is not sufficient for publishing, and therefore I recommend rejection.